# Effect of easing lockdown and restriction measures on COVID-19 epidemic projection: A case study of Saudi Arabia

**Shafiun Nahin Shimul**[1]*, **Angi Alradie-Mohamed**[2], **Russell Kabir**[2], **Abdulrahman Al-Mohaimeed**[3], **Ilias Mahmud**[4]

**1** Institute of Health Economics, University of Dhaka, Dhaka, Bangladesh, **2** School of Allied Health, Anglia Ruskin University, Essex, United Kingdom, **3** Department of Family and Community Medicine, College of Medicine, Qassim University, Buraydah, Saudi Arabia, **4** Department of Public Health, College of Public Health and Health Informatics, Qassim University, Al Bukairiyah, Buraydah, Saudi Arabia

* shafiun@huskers.unl.edu, shafiun.ihe@du.ac.bd

**Data Availability Statement:** The data are publicly available through Dryad (DOI: 10.5061/dryad.p8cz8w9r6).

## Abstract

### Objectives

In this study we compared two predictions of COVID-19 cases in the Kingdom Saudi Arabia (KSA) using pre–and post–relaxation of lockdown period data to provide an insight regarding rational exit strategies. We also applied these projections to understand economic costs versus health benefit of lockdown measures.

### Methods

We analyzed open access data on COVID-19 cases from March 6 to January 16, 2021 in the KSA. To understand the epidemic projection during the pre- and post-lockdown period, we used two types of modeling: the SIR model, and the time series model. We also estimated the costs and benefits of lockdown- QALY gained compared to the costs of lockdown considering the payment threshold of the Government.

### Results

Prediction using lockdown period data suggested that the epidemic might slow down significantly after 109 thousand cases and end on October 6, 2020. However, analysis with latest data after easing lockdown measures suggested that epidemic might be close to an end on October 28, 2021 with 358 thousand cases. The peak has also been shifted from May 18, 2020 to Jun 24, 2020. While earlier model predicted a steady growth in mid-June, the revised model with latest data predicted it in mid-August. In addition, we estimated that 4986 lives would have been saved if lockdown continued but the cost per life saved would be more than $378 thousand, which is way above not only the KSA threshold, but also the threshold of any other highly advanced economies such as the UK and the USA.

### Conclusions

Our results suggest that relaxation of lockdown measures negatively impacts the epidemic. However, considering the negative impact of prolong lockdown measures on health and

**Funding:** The author(s) received no specific funding for this work.

**Competing interests:** The authors have declared that no competing interests exist.

economy, countries must decide on the best timing and strategy to exit from such measures to safely return to normal life with minimum loss of lives and economy considering its economic and health systems' capacity. Instead of focusing only on health, a balanced approach taking economy under consideration is recommended.

## Introduction

Corona Virus Disease 2019 (COVID-19) is a newly discovered infectious disease of the respiratory system which is caused by the Severe Acute Respiratory Syndrome Coronavirus 2 (SARS-Cov-2) [1]. COVID-19 was first reported in Wuhan, China [2,3], and ravaged the world to become a pandemic [4]. It is highly infectious with an incubation period of 2–14 days. An infected case can spread the disease even in the time between suspecting the case and confirmation of the case [5]. This makes it difficult to suppress the speed of the transmission of the disease.

As of January 18, 2021, globally there have been 93,611,355 confirmed cases of COVID-19 with 2,022,405 deaths (World Health Organization (WHO, 2020). In the Kingdom of Saudi Arabia (KSA), the first case of COVID-19 was reported on March 2, 2020 [6–8]. Since then, as of January 18, 2021, the epidemic of COVID-19 has spread over 206 cities and infected 3,65,099 people and caused 6329 deaths in the KSA [6].

COVID-19 therapeutic strategies are only supportive [9], thus prevention of transmission of the disease in the community is the best strategy. KSA started implementing early preventive measures even before any case was recorded in the country [8]. To contain the COVID-19 epidemic locally and globally and to limit human and economic losses, the KSA took one of the strictest approaches [7]. The measures taken by the KSA included, but not limited to, closing the international borders, closing the two grand mosques in Mecca and Medina for both national and international religious tourists closing all mosques for prayers including Friday prayers, closing inter-regional public and private transports including air travel, closing down all shopping malls, gymnasiums, and other public recreational facilities, closing all public offices apart from vital service providers, closing all educational institutions and moving to online teaching. In addition, government also enforced a few more steps such as imposing partial to complete lockdown of cities depending on its outbreak situation, prohibiting mass gatherings even in private places such as home, imposing all public health measures recommended by the WHO and the Saudi Ministry of Health (MoH) with heavy penalty for violation. Along with these restrictions measures the MoH dedicated 25 hospitals for COVID-19 patients with 80,000 hospital beds and 8000 Intensive Care Unit (ICU) facilities, did mass testing, contact tracing and mandatory medical isolation of suspected and confirmed cases [7,8]. Although these measures could not prevent the disease of becoming an epidemic within the country in the long-run, the early implementation of precautionary measures enabled KSA to slowdown the spread of the virus, hence provided time to increase awareness and promote a culture of adhering to the prescribed prevention measures among the residents. It also enabled KSA to maintain its health system and double the number of laboratories, intensive care units and ventilators in the hospitals [10].

Although scientists fear a second wave of COVID-19 outbreak after relaxation of lockdown measures [11], many questioned the viability of such measures for a longer time citing its negative effect on all sectors including health, education and economy [7]. On May 25, 2020, the health minister of the KSA declared a two-pronged strategy to return to normal life [10]. The

strategy involved further strengthening the capacity of hospitals to serve critical cases and intensifying mass testing to detect infected cases early. Starting from May 29, 20020, KSA focused on social distancing, and gradually eased lockdown measures. COVID-19 curfew and lockdown measures were completely lifted on June 20, 2020. However, educational institutions continued to operate online, and government offices were instructed to continue to perform online when physical presence was not necessary.

Lockdown and other strict measures are taken to suppress the infection but these cannot be considered as long-term measures. Closures of the economic activities affects the economy significantly, and this number may go up to 40% of the GDP. Lockdown and its impact on infection before and after lockdown measures were studied in many recent literature [12–15] with various modeling framework with some modifications of the Susceptible-Infected-Recovered (SIR) model. In addition, some literature also applied econometric techniques to estimate the impact of easing or enforcing lockdown such as Ibrahim et al and Ajide [16,17]. On the other hand, a handful of literature observed the economic costs of lockdown [18]. For instance,Harvant et al, 2020 show that even though there in negative shocks in the GDP due to lockdown, the impact will be heterogeneous across sectors. In addition, effects on lockdown on other sector such as air-quality is also found in the literature [19,20]. However, while lockdown can help suppress infection and save lives, it also wreaks havoc to the economy, both of which should be considered for decision making.

Several studies used the SIR model to make predictions regarding the development of COVID-19 in different countries and evaluate the effect of lockdown and prevention measures [21,22]. Using the SIR model in KSA, a study predicted the peak of the epidemic in the KSA on May 1, with the steady phase beginning on June 2, and the ending phase starting on June 24, 2020 [23]. The use of SEIR model is also evident to understand the impact of lockdown [24]. However, their prediction was made before easing the restriction measures. In this article we compared several predictions using the SIR model- one before relaxation of lockdown measures and the others after the relaxation. This research could provide useful information regarding the best timing and strategy to exit from any future similar infectious disease epidemic restriction measures to safely return to normal life with minimum loss of lives and economy. Although this study primarily compared the number infection before and after relaxation of the lockdown measures in the KSA, based on the evidence available we also explored the economic costs and benefits of such intervention. Then provided few recommendations based on the cost-benefit analysis. Lessons learned from the KSA could also be applied to other countries to adopt an effective exit strategy from lockdown and other strict restriction measures.

## Materials and methods

We used open access data from the online database of the Our World in Data, a project of the Global Change Data Lab [25]. For analysis, we used data from March 6, 2020 to Jan 16, 2021. While to capture pre-lockdown prediction March 6, 2020 to May 30, 2020 data were used, for post-lockdown prediction data until January 16, 2021 was used with break points on July 1, 2020, August 5, 2020 and January 16, 2021. To understand the spread of the virus and the impact of various policies adopted by the KSA, we used two types of modeling: the SIR model, and the time series model. The SIR model is a compartmental model of infectious disease widely used for predicting and understanding COVID-19, developed by Kermack, and McKendrick [26]. It is basically solving a set of differential equations:

$$\frac{dS}{dt} = -\beta \frac{SI}{N} \tag{1}$$

$$\frac{dI}{dt} = \beta \frac{SI}{N} - \gamma I \qquad (2)$$

$$\frac{dR}{dt} = \gamma I \qquad (3)$$

where $\beta$ represents transmission rate and $\gamma$ represents the rate of recovery or death among infected respectively, and R+S+ I = N, where N is population, which is assumed to be constant.

Eq (1) shows changes in S which is inversely related to number of people infected and transmission rates. Eq (2) provides changes in $I$ which is the difference between number of infections and recovery. Finally, Eq (3) describes change in recovery or death and so removed from the system. Solving these equations with some initial assumptions, and with some set of constraints, we can estimate basic reproduction rate/number, $R_0 = \beta/\gamma$, where $\beta/\gamma$ represents contact ratio, and $S_o$ is the initial susceptible number of population. This ratio shows the number of new infections from a single infection in a population where all subjects are susceptible. In other words, $R_o$ denotes the infectiousness of the diseases—with higher $R_o$ denoting higher infection capability. If $R_0$ value of greater than 1, then the epidemic will be worsening very quickly. Similarly, we can also estimate effective reproduction number, $R_t$. While $R_o$ provides an estimate of infectiousness of the pathogen, $R_t$ provides infection level (rate) at certain point in time, and therefore, $R_t$ is often is used to adopt and see the impact of various policies. $R_t$ also provides information on how the infection is spreading—with greater than 1 indicating exponential growth and less than 1 indicating decay in growth.

There are two ways we can estimate the impact using the SIR model. One is using simulation with the assumption of parameters. When we do not have any data, this is the only option. In this case, we take the parameters from the literature and from the other countries and then simulate. For instance, if we have the data of infection rate, contact rate, and other parameters, we can simulate the model to understand the various infection levels under various policy scenarios. While this tool is useful, it is highly sensitive to the value of the parameters and so the actual number may turn out to be significantly higher or lower than the predicted numbers. This is what happened in the case of a study [27] which estimated that in the USA the actual infection will be several millions with 2.2 million deaths. Although the epidemic has not yet ended, it is very less likely that the actual number will be even close to what was predicted.

Second way to estimate parameters is using numerical solutions, i.e., solving models from the data. We followed this route for various reasons. First, we have the data in hand- good enough to run numerical optimization. Second, instead of assuming other countries' parameters, we can estimate parameters for the country under question- so context is utilized and therefore, more realistic. For numerical solutions, a enough data is needed. Fortunately, we have that for COVID-19. The amount of data we have is good enough to run the numerical SIR model.

In SIR model, first, we get the parameters from the logistic fits of the curve as the epidemic curve fit logistic distribution quite well, and these parameters are used as an initial guess for the differential equation. Then we estimate the parameters of the model with this numerical solution:

Step 1: set up the model

Step 2: estimate the parameters of logistic functions

Step 3: use the parameters of logistic function as initial solution

Step 4: estimate the parameters of the SIR model using the parameters of initial value (from logistic function)

Step 5: predict *S*, *I*, *R* and then all other key dates; most importantly, estimation of $R_o$ and $R_t$.

The value of $R_t$ is what is important in our case.

We estimated *R* before and after lifting the lockdown and restriction measures. Since COVID-19 has a maximum of 14 days of incubation period, we considered these periods as transition periods. When KSA lifted the lockdown, the number of infections were going down, and so it was expected that R will be even lower. Whether that is the cases to be seen using the differences in the expected and actual value of R and the infected cases. We estimated I and R before and after lifting the lockdown and relaxing other restriction measures and then compared them to understand the impact of lifting the lockdown and relaxing other restriction measures.

Although the SIR model is a widely applied epidemiological model, it is restrictive in term of assumptions of parameters. In contrast, the time series model is more flexible. We used the Gompertz distribution since this distribution allows long tail unlike other commonly used distribution such as the Logistics. The Gompertz model [28]–a sigmoidal model–is widely used for growth and other similar data. Its use spans from plant growths, animal growths to growth of pathogens. It is also used in calculating survival and death. The flexible nature of this model makes it a natural choice in understanding COVID-19 projections.

The simplified version of the Gompertz curve can be expressed as:

$$I_t = Be^{e^{-\beta(t-K)}} \tag{4}$$

$I_t$ is the cumulative cases at time t, B is the upper asymptote. $\beta$ is the growth rate of infection, *K* is the point of inflection. The cumulative cases are estimated through three parameters: B and $\beta$, K, and applying a non-linear regression. Once we estimate the parameters with non-linear regression, we can predict $I_t$ which is infection cases. Higher $\beta$ indicates higher growth of infection.

Positive impact of lockdown and other strict measures on health are given priority in deciding in favour of these measures. However, these measures negatively affect the economy, and this loss is often ignored. A recent study [18] showed that the gain achieved through lockdown in comparison with other health interventions in the UK is very costly. For instance, in the UK, £20,000 –£30,000 is used as the threshold costs for each Quality Adjusted Life Years (QALY) gains from any health intervention. If lockdown is compared with no-lockdown to understand the real benefit (QALY gains) versus loss in economy (% in GDP loss), we can understand that the cost of lockdown far outweighs the benefit [18].

Although there is no QALY threshold estimated for the KSA, it can be assumed that QALY threshold of the KSA will be much lower compared to that of the UK as per capita income of the KSA is almost half compared to the UK. If we use QALY threshold of the UK in proportion to the income and use the same ratio for the KSA, we find that the threshold should be 10,000– 17,000 USD. We first estimated the incremental benefit of lockdown in terms of QALY gains —i.e. we estimated QALY gain with lockdown and QALY gain without lockdown and then computed the difference. Then, we estimated loss in GDP due to lockdown measures. Once the loss is estimated, we can estimate the cost for each QALY gain. Finally, we compared the cost per QALY gain with maximum threshold to understand the justification of lifting or keeping lockdown measures. The detailed methodologies are below:

Lives saved due to lockdown: we compared potential deaths with lockdown and without lockdown. Since our projection models would provide information on total cases, we can estimate the difference in cases with and without lockdown. Total deaths are estimated using the COVID-19 case fatality rate of the KSA. Then lives saved were computed by subtracting the

average age of the people died of COVID-19 in the KSA (47.1 years) from the average life expectancy in the KSA which is 74.13 (World Development Indicators, 2021, World Bank).

Then we converted this death number and other co-morbidity (among those who survived) into QALY lost. [29] states that on an average 1.5 QALY for each infection. However, Miles [16] assumes it as 50% of death numbers as majority of the deceased persons had the history of other morbidities and so they had lower QALY in the first place. In this study, we assumed 1.5 QALY loss for each infection. Then, GDP loss for the time frame of lockdown is estimated assuming various levels of disruption in economic activities. Finally cost per QALY gain is estimated to compare this ration with potential threshold.

## Results

We applied both the SIR and Gompertz models to understand the lockdown and restrictions related policy impacts on the COVID-19 epidemic in the KSA. Tables 1 and 2, in the appendix, show the changes in estimated parameters, projected cases before and after easing lockdown and restrictions measures in the KSA. Figs 1–3 provides epidemic curves under two settings and Fig 4 shows actual reported cases in the KSA. To see the immediate impact, parameters estimated with data from March 6, 2020 to July 1, 2020 were compared with estimated parameters before easing lockdown (March 6 to May 30, 2020). Before easing the lockdown measures the effective reproduction number was less than .94, exhibiting a downward trend in the growth of infection. However, after easing the lockdown measures, effective reproduction number has increased to .97. This changes in the effective reproduction rate demonstrates that easing lockdown has increased the infection potential. This finding is further substantiated by increase in time between contacts (Tc) and infectious period (Tr).

While prediction using the lockdown period data suggested that epidemic may slow down significantly after 109 thousand cases. Immediately after easing lockdown, the prediction

**Table 1. SIR model predicting COVID-19 epidemic in KSA in lockdown and post-lockdown scenarios.**

| Estimate SIR Model Parameters | Pre (Until May30) | Post (Until July1) | Post (Until Aug 5) | Post (Until Jan 16, 2021) |
|---|---|---|---|---|
| Contact frequency (beta) | 0.441 | 0.372 | 0.268 | 0.072 |
| Removal frequency (gamma) | 0.358 | 0.323 | 0.22 | 0.026 |
| Basic reproduction number (Ro) | 1.231 | 1.151 | 1.218 | 2.8 |
| Effective reproduction number (Rt) | 0.936 | 0.971 | 0.861 | 0.22 |
| Time between contacts (Tc) | 2.3 | 2.7 | 3.7 | 13.9 |
| Infectious period (Tr) | 2.8 | 3.1 | 4.6 | 38.8 |
| Epidemic size (maximum infection) | 108720 | 299707 | 327097 | 358031 |
| Epidemic rate (percent of population) | 0.0827682 | 0.048709 | 0.0478195 | 0.0462562 |
| Total epidemic duration (in days) | 236 | 386 | 408 | 679 |
| Outbreak | 6-Mar-20 | 6-Mar-20 | 6-Mar-20 | 06-Mar-2020 |
| Start of acceleration | 25-Apr-20 | 9-May-20 | 22-Jun-20 | 11-May-2020 |
| Turning point (peak) | 19-May-20 | 19-Jun-20 | 6-Aug-20 | 24-Jun-2020 |
| Start of steady growth | 15-Jun-20 | 1-Aug-20 | 19-Sep-20 | 16-Aug-2020 |
| Start of ending phase | 10-Jul-20 | 13-Sep-20 | 12-Mar-21 | 04-Oct-2020 |
| End of epidemic (5 case) | 6-Oct-20 | 20-Feb-21 | 11-May-21 | 28-Oct-2021 |
| RMSE | 769.885 | 3420.54 | 3140.48 | 4761.64 |
| R-sq | 0.999 | 0.997 | .999 | 0.999 |

Note: Contact frequency (beta): Higher the frequency higher the contact, higher the infection, other thing remaining constant.

**Table 2. Gompertz distribution: Parameter estimation (before and after lockdown).**

| | (1) | (2) | (3) |
|---|---|---|---|
| | **During Lockdown** | **Immediately After Lockdown** | **Few months after Lockdown** |
| $B$ (upper asymptote) | 257405.67*** | 434546.42*** | 429394.34*** |
| | (11167.77) | (7816.18) | (7521.98) |
| $\beta$(growth rate of infection) | 0.0269*** | 0.0212*** | 0.0214*** |
| | (41.52) | (52.58) | (52.69) |
| K (point of inflection) | 22069.27*** | 22088.76*** | 22088.16*** |
| | (1.46) | (0.97) | (0.95) |
| Log-likelihood | -669.04 | -1470.70 | -1483.81 |
| N | 86 | 153 | 154 |

Standard errors in parentheses.

\* $p < 0.05$

\*\* $p < 0.01$

\*\*\* $p < 0.001$.

$I_t$ is the cumulative cases at time t, B is the upper asymptote. $\beta$ is the growth rate of infection, $K$ is the point of inflection. The cumulative cases are estimated through three parameters: B and $\beta$, K, and applying a non-linear regression. Once we estimate the parameters with non-linear regression, we can predict $I_t$ which is infection cases. Higher $\beta$ indicates higher growth of infection.

suggests that epidemic will be close to end after 300 thousand cases, which is substantially different than the previous projected numbers, and if latest data is considered the tally becomes even bigger (358 thousand cases). The shift of the peak from mid-May to early August is also observed. The SIR model run after lockdown demonstrates that the previous peak was not necessarily the peak. While earlier model predicted a steady growth in mid-June, later models suggest a shift to mid-September/August. Moreover, the start of ending phase—when curve will be plateaued—has also shifted significantly. The latest model predict that epidemic may not

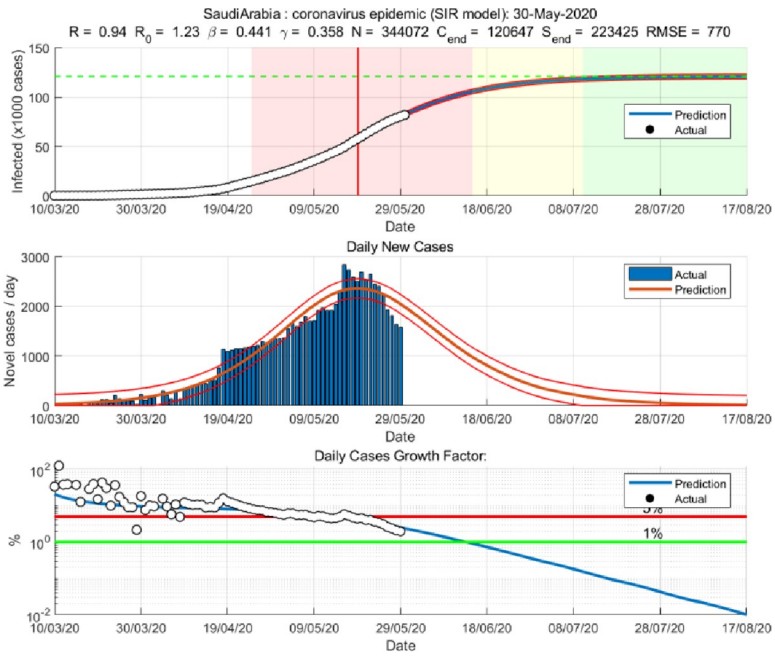

**Fig 1. SIR model during lockdown period (until May 30, 2020).**

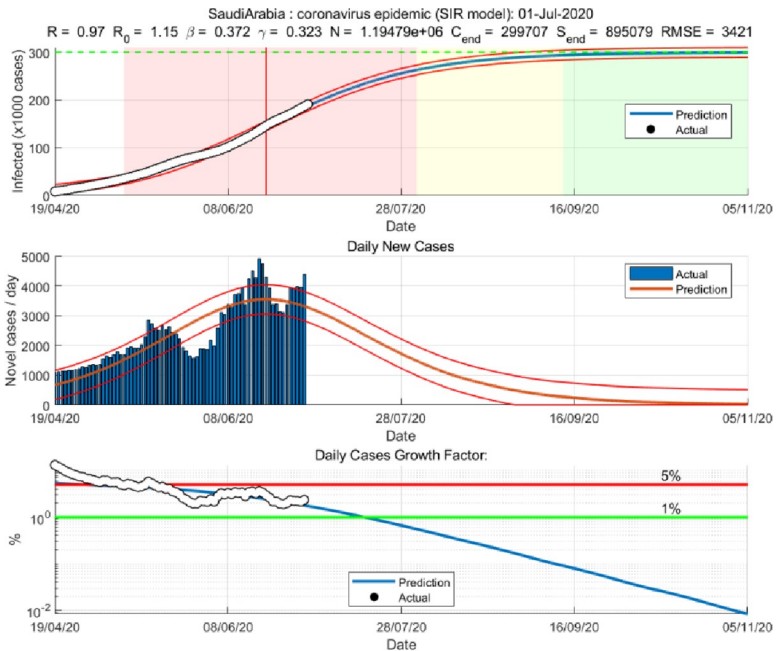

**Fig 2. SIR model based on post- lockdown data (until July 1, 2020).**

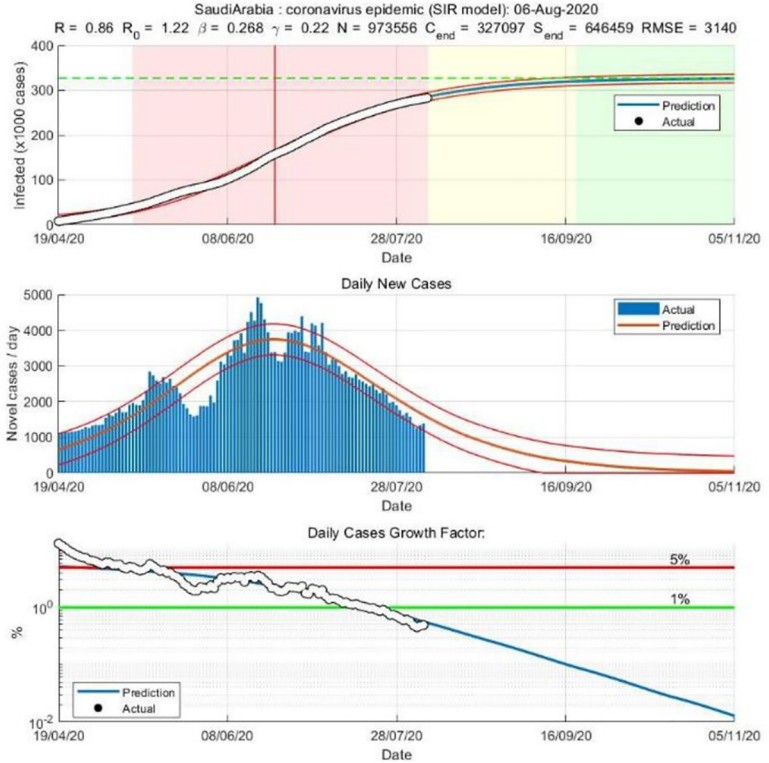

**Fig 3. SIR model based on post-lockdown data (until Aug 5, 2020).**

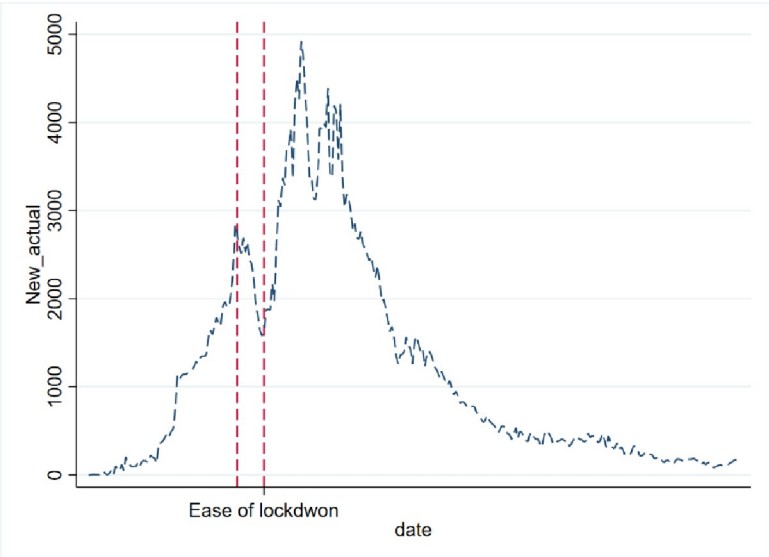

**Fig 4. Actual daily COVID-19 new cases in KSA.**

end until last quarter of 2021, though the previous model predicted an early end of the epidemic. Interestingly, the explanatory power of the SIR models (in both pre and post) remains very high—at 99%, which mean 99% of the variation in the data can be explained by the model.

In addition, the Fig 5 also shows how the predication and actual cases changed dramatically after 14 days of easing lockdown and restriction measures. From the figure it is observed that while predicated and actual values were going hand-in-hand until May 30, actual cases and the predicated cases (with data from March 6, 2020 to Aug 5, 2020), has had a drastic change.

We also estimated the parameters of Gompertz distribution using non-linear regression. Table 1 reports the parameter estimates, it is observed that growth rate of infection (β) has

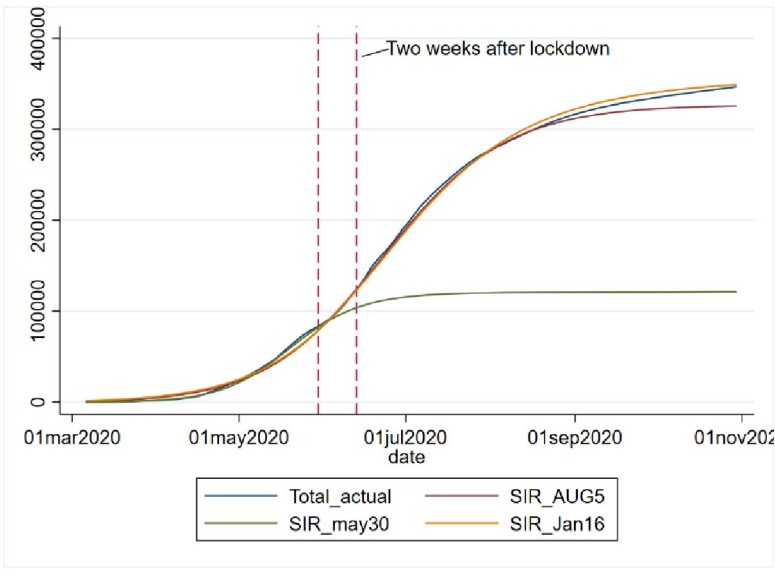

**Fig 5. Impact of easing lockdown measures on COVID-19 epidemic in KSA.**

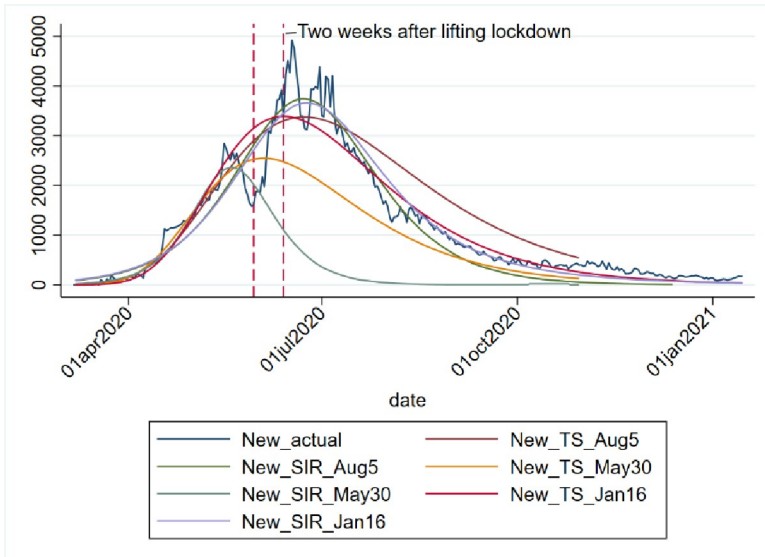

**Fig 6. Projection of daily COVID-19 cases in KSA under various scenario.**

increased after relaxing lockdown and the K (max asymptotic cases) has increased as well and both estimates are statistically significant (p-value<0.05). However, the inflection point remains very much same.

With projected values estimated from non-linear regression with the Gompertz distributional assumption, we can compare the impact of policy changes by looking at the differences in the predicated values and their comparison with data. The Fig 6 shows the infection curves of actual total cases, predicted total cases from the regression run with data until lockdown in place and afterwards. While forecast with data of March 6, 2020 to May 30, 2020 was total over 120,000 cases by the end of September, the same model when run with newer data, as shown in the Fig 7 from March 6, 2020 to Jan 16, 2021 predicts close to 400 thousand cases in the KSA. Hence, we see that easing lockdown resulted in higher infection—which was demonstrated through both the SIR and Gompertz model. Furthermore, the Gompertz model also predicts that the peak will be shifted (see the graphs in appendix).

For KSA, we estimated that 4986 lives would have been saved if lockdown continued but the cost per life saved is estimated at more than 378 thousand US dollar, which is way above not only the KSA threshold, but also threshold of any highly advanced economies such as the UK and the USA (the USA threshold is $100,000-$150,000). However, the loss in GDP would determine the cost per QALY gain for lockdown. As shown in the Table 3, even if we assume 10% loss in GDP, then cost per QALY gain is also very high.

## Discussion

We used both the SIR and the Gompertz distribution model to understand the policy impact in KSA. While the SIR model provided some epidemiological parameters, the Gompertz model better demonstrated the changes in predicated COVID-19 cases for the KSA. This study suggests that the KSA has eased the curfew perhaps little early in terms of infection spread potential. Even though daily new cases were on the decline and $R_t$ was much lower than 1, things did not go the way it had been expected. For example, in March 2020, a study using SIR model predicted that COVID-19 outbreak will begin the final-phase in the KSA by end of June 2020 [23], while another study [30] predicted using the SIR model that the disease will

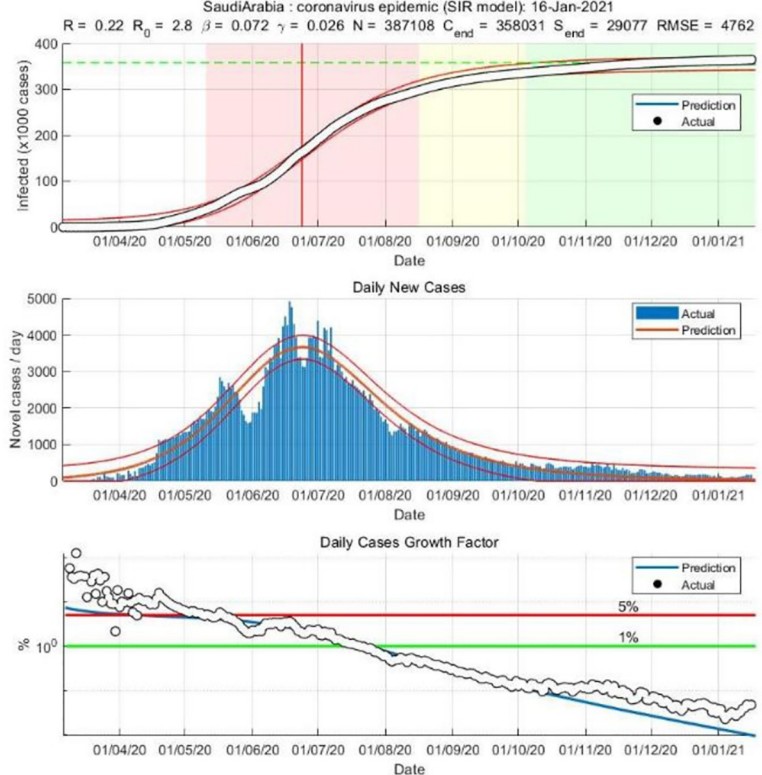

**Fig 7. SIR model based on post-lockdown data (until January 16, 2021).**

peak in the Mecca city on 22nd of May, and then the number of cases will decrease. The results of our study reflect how easing the lockdown measures negatively affected those predictions. This study results shows that the value of Rt is less than one, indicating the decline in trend of new infections. That means using classic epidemiological decision tool -Rt must be less than 1-, therefore, it was not unscientific for the KSA to relax restrictions. However, the situation has worsened since then. The update in the situation needs further investigation and discussion.

Firstly, before 30th May Saudi's peak was on 17th May, two weeks before easing lockdown and restriction measures, and so everything was as expected—Rt was less than 1, new cases were falling constantly. The 7th of May was the national peak date, however, putting in consideration that the KSA is a big country; some parts of the country might have not reached the peak by that time. Therefore, in deciding lifting lockdown, the geographical distribution of cases should have been considered as well.

Secondly, it has been observed that many countries experienced more than one peak such as the USA, UK, India and France [25]. Therefore, if a peak is observed it could be just a natural fluctuation of data, instead of a peak. Until the daily new cases reach in an extremely low level, it is perhaps desirable to be extra careful, otherwise, things may take an unexpected turn.

**Table 3. Costs per QALY gain for lockdown with various levels of GDP loss due to lockdown.**

| GPD loss (%) | 10% | 20% | 30% | 40% | 50% |
|---|---|---|---|---|---|
| Cost per QALY gain | 94,557 | 189,114 | 283,671 | 378,228 | 472,786 |

Thirdly, since there remains an insurmountable amount uncertainty regarding the COVID-19, the policymakers need to consult with various modeling framework. Along with the SIR model—a classical epidemiological model—other models deserve attention for better understanding of the potential epidemic trajectories. A restrictive version of the SIR model assumes equal tails of the distributions; however, based on many countries' experiences, it turns out that for COVID-19, this is not necessarily the case. Even the countries that were successful in containment, they had long tail in the right-hand side [25]. The Gompertz type distribution can capture that phenomenon.

When comparing between the value of Rt pre and post relaxation of restrictions, the value of Rt was less than .94 meaning downward trend in the growth of infection, then the value of Rt increased to .97 meaning increase in the potentiality of infection. This can be interpreted as to how effective the restrictions were in decreasing the growth of the disease. Furthermore, compared to the pre-relaxation period in the post-relaxation period the time between contacts Tc and infectious period Tr have also increased.

Data analysis using the SIR and Gompertz models predict the shift in the peak from mid-May to early August 2020. Findings of this research suggest that the previous observed peak was not the actual peak and predicts a steady growth in mid-September 2020. However, a study [31] reported a predicted peak on 13th of July 2020 and justified observing double peaks behavior in the KSA by removal and reapplication of lockdown within a short period.

Another study conducted in the KSA [32] used the ARIMA model and Logistic growth model to provide short and long-term COVID-19 predictions; their prediction were made assuming that the lockdown will remain, KSA residents would follow the recommended safety guidelines and non-pharmaceutical interventions would be maintained. Based on that assumptions they predicted the end of the epidemic by 5th of August in their first scenario. However, like our study they also observed increase in cases upon temporarily relaxation of curfew in May. Nevertheless, they still predicted the end of the epidemic on 29th of September [32], based on the same assumptions. However, their scenarios were disturbed as proven by our study, due to easing lockdown measures.

Our study results disprove the prediction made by a study conducted in March, regarding the peak of the epidemic, start of ending period of the epidemic, and their suggestion that warm weather may contribute to decreasing the spread of the disease [23]. This study also contradicts another study [33] which predicted that the epidemic would peak early on 27th of March, and the end of epidemic phase would be 18th of April 2020. Khoj and Mujallad [34] had similar predictions to that of Alnoaneen et. al. [23] and Komies et. al. [33]. The contradictions between our study and these studies can be justified as the data used in these studies belonged to the time before easing of lockdown and restrictions measures.

We used two models: the SIR and Gompertz for two sets of data. This research is the first of its kind in the KSA to compare COVID-19 epidemic predictions between pre–and post–relaxation of lockdown measures phase. This study is important as it shows how changes in policies like easing lockdown can affect the spread of the disease.

The explanatory power of the SIR models using the pre–and post–relaxation of lockdown is very high (99%). However, the findings of this study should be interpreted with caution. Firstly, the models under study did not consider age-related contact rate- immunity and fatality from COVID-19 differ across age groups, or rates of underreported infections [35]. We also need to acknowledge that our estimations are based on the reported cases which is very likely to be underestimated. Our data included reported cases including those who visited hospitals or were tested for other reasons but not included those who had symptoms or were asymptomatic and were not tested. Under-reporting and asymptomatic cases were observed in many countries around the world, and it can cause under-estimation of the accumulated cases

[23]. On the other hand, around mid-April 2020, KSA started to implement active surveillance program-testing COVID-19- among several communities on different dates [8], the testing include testing everyone in the assigned community–symptomatic and asymptomatic- this can lead to some extend increase of recorded cases at some points in the timeline, and over estimation when compared to times where mass testing was not preformed.

Furthermore, other factors can affect the predictions of this study; for example, if new restriction measures were implemented, or opening of international airports can shift both the peak and the ending date of the epidemic. As stated in a study using the Susceptible-Exposed-Infectious-Recovered (SEIR) that the accurate SEIR prediction will only be obtained after COVID-19 outbreak has been successfully been controlled [36].

It is worth mentioning that official declaration of lockdown is used in this study; however, citizen's ultimate compliance can probably better explain the impact of lockdown on the growth infection. Hence, use of google mobility data as a proxy for lockdown could also be applied as done in [37]. However, it is beyond the scope of current study.

Based on the findings of this study it is recommended to return the country to some sort of restriction, to lower the potentiality of higher infection, as it appears that lockdown and movement restrictions had positive effect in controlling COVID-19 epidemic in the KSA and since the national level lockdown can be costly in terms of economic shutdown, sub-national level lockdown can be considered. Moreover, as lockdown cannot be a long-run solution, we suggest national health education campaign with specific target to particularly vulnerable population–low economic areas, crowded residential areas. Increase knowledge of COVID-19 and sharing the responsibility with the population might decrease the transmission of the virus [38].

Our results suggest that in case countries decide not to return to lockdown, the country should prepare for long COVID-19 period and prepare heath facilities for more COVID-19 cases, while providing education campaigns. Since there remains an enormous amount uncertainty regarding the COVID-19 trajectory, various modeling frameworks need to be consulted to better capture possible range of paths of the epidemic. Using established model such as the SIR and other time series models, this study finds that lifting lockdown has increased in the infection in the KSA. However, as lockdown is not feasible option for long-term, other public health measures can be adopted. Even that policy would result higher infection and deaths, considering distorted normal life and enormous loss to the economy, lockdown cannot continue for long. However, a country should decide an optimal time of lifting lockdown and geographical variation need to be considered.

We showed that if we look at the R and decide based on that we may be misguided with regards to when the pandemic will end as the Ro estimated during lockdown will not remain same if lifted lockdown too early. This is exactly what happened to the KSA. However, when Ro is way below one, not just close to one, say at best 0.80, then economy may open. As waiting for Ro go down too much may leave high economic loss. In addition, using cost-benefit analysis it appears that keeping restriction for long time when R is way below 1 does not make economic sense. While this is true for the KSA, to make this result generalizable to other country, one must consider that country context. This study focused on national level measures. Future research can develop an optimal policy with geographically disaggregated data.

## Author Contributions

**Conceptualization:** Shafiun Nahin Shimul, Ilias Mahmud.

**Data curation:** Shafiun Nahin Shimul.

**Formal analysis:** Shafiun Nahin Shimul, Ilias Mahmud.

**Investigation:** Abdulrahman Al-Mohaimeed.

**Methodology:** Shafiun Nahin Shimul, Angi Alradie-Mohamed, Russell Kabir, Abdulrahman Al-Mohaimeed.

**Validation:** Russell Kabir.

**Visualization:** Shafiun Nahin Shimul.

**Writing – original draft:** Shafiun Nahin Shimul, Ilias Mahmud.

**Writing – review & editing:** Shafiun Nahin Shimul, Angi Alradie-Mohamed, Russell Kabir, Abdulrahman Al-Mohaimeed, Ilias Mahmud.

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
