## [Decision Letter · Decision Letter 0]

19 Mar 2021

PONE-D-21-01867

Effect of easing lockdown and restriction measures on COVID-19 epidemic projection: a case study of Saudi Arabia

PLOS ONE

Dear Dr. Shimul,

Thank you for submitting your manuscript to PLOS ONE. After careful consideration, we feel that it has merit but does not fully meet PLOS ONE’s publication criteria as it currently stands. Therefore, we invite you to submit a revised version of the manuscript that addresses the points raised during the review process.

In addition to the technical comments by Reviewer #1, please pay special attention the requests by Reviewer #1 and Reviewer #2 on:

reporting parameters obtained from fitting the epidemic curve to the Gompertz curve, along with uncertainties;explaining how this work can help identify the correct time for easing lockdowns; andshowing some form of cost-benefit analysis.

We look forward to receiving your revised manuscript.

Kind regards,

Siew Ann Cheong, Ph.D.

Academic Editor

PLOS ONE

Journal Requirements:

4. Please ensure that you refer to Figure 5 in your text as, if accepted, production will need this reference to link the reader to the figure.

5. Please include captions for your Supporting Information files (SI 2) at the end of your manuscript, and update any in-text citations to match accordingly. Please see our Supporting Information guidelines for more information: http://journals.plos.org/plosone/s/supporting-information.

Reviewers' comments:

Reviewer's Responses to Questions

**Comments to the Author**

1. Is the manuscript technically sound, and do the data support the conclusions?

Reviewer #1: Partly

Reviewer #2: Yes

2. Has the statistical analysis been performed appropriately and rigorously? 

Reviewer #1: Yes

Reviewer #2: Yes

3. Have the authors made all data underlying the findings in their manuscript fully available?

Reviewer #1: Yes

Reviewer #2: Yes

4. Is the manuscript presented in an intelligible fashion and written in standard English?

Reviewer #1: Yes

Reviewer #2: No

5. Review Comments to the Author

Reviewer #1: The authors use two mathematical models, SIR (compartment type) and Gompertz (time series) models, with curve fitting methods to describe the time evolution of the COVID-19 pandemic in the Kingdom Saudi Arabia before and after lifting the lockdown. It is found that predictions made before the lifting of the lockdown are not accurate (too optimistic) and the predictions made after lifting the lockdown matches real data better and with longer duration of the pandemic. The results suggest that relaxation of lockdown measures negatively impact the epidemic. Judging from the negative impact on the economy due to lockdown, the authors suggest a country should decide an optimal time of lifting lockdown and

geographical variation need to be considered.

Similar methods have been employed in analyzing the COVID-19 pandemic in existing literature during and after the lockdown. See, for example: Science 369, Issue 6500, eabb9789; Eur Phys J Plus. 2020; 135(11): 885; CJP vol:70, p170, 2021; Scientific Reports (2020) 10:22454, etc., just to name a few. Thus the novelty of utilizing data before and after the lockdown is not that apparent. There are also several flaws in the current manuscript needed to be addressed (listed below). Thus I cannot recommend its publication in the PLOS One.

1. Equation (1) and (2) is not correct. A factor of 1/N is missing.

2. The basic reproduction number Ro is not correctly defined ("So" mentioned in Ro's definition should be the initial

susceptible number. This is not defined in the manuscript either.). The proper definition should be defined by So/N,

not just So. The effective reproduction number Rt is not defined nor properly referenced.

3. Similar to point 2, there are several parameters/terminologies in Table 1 are not defined.

4. All figures have no captions.

5. Ref. 15 and 17 are missing.

6. Results of time series are shown only in Fig.6, and the fittings are not as good as SIR ones. What's the point of using

time series analysis?

7. Equation of Gompertz curve is mentioned on page 5, but the parameters for obtaining the fitting in the Fig.6 are not

mentioned in the manuscript.

8. The comparison with I and R are not shown. Fig.1-6 are just daily new cases=-dS/dt and/or infected=N-S.

9. The negative impact on the pandemic for early lifting of the lockdown seems obvious even without mathematical

modeling. How can this work help identify the proper timing for easing the lockdown?

Reviewer #2: Is the manuscript presented in an intelligible fashion and written in standard English?:

The paper needs to be read and amended by someone with English as there first language

This interesting and thought provoking paper addresses a complex issue which is bounded by uncertainty. It is helpful to have a perspective from Saudi Arabia on this global issue.

The major thing that needs to be addressed is that the results section narrative needs to have much more detail in relation to what is shown in the figures.

There are also economic considerations which have to be factored in as described in: Miles DK, Stedman M, Heald AH. "Stay at Home, Protect the National Health Service, Save Lives": A cost benefit analysis of the lockdown in the United Kingdom. Int J Clin Pract. 2020

Please can this cost benefit analysis be factored in

6. PLOS authors have the option to publish the peer review history of their article (what does this mean?). If published, this will include your full peer review and any attached files.

Reviewer #1: No

Reviewer #2: **Yes: **Adrain Heald

---

## [Decision Letter · Decision Letter 1]

29 Jun 2021

PONE-D-21-01867R1

Effect of easing lockdown and restriction measures on COVID-19 epidemic projection: a case study of Saudi Arabia

PLOS ONE

Dear Dr. Shimul,

Thank you for submitting your manuscript to PLOS ONE. After careful consideration, we feel that it has merit but does not fully meet PLOS ONE’s publication criteria as it currently stands. Therefore, we invite you to submit a revised version of the manuscript that addresses the points raised during the review process.

In particular, please address the comments made by Reviewer 3.

We look forward to receiving your revised manuscript.

Kind regards,

Siew Ann Cheong, Ph.D.

Academic Editor

PLOS ONE

Journal Requirements:

Reviewers' comments:

Reviewer's Responses to Questions

**Comments to the Author**

1. If the authors have adequately addressed your comments raised in a previous round of review and you feel that this manuscript is now acceptable for publication, you may indicate that here to bypass the “Comments to the Author” section, enter your conflict of interest statement in the “Confidential to Editor” section, and submit your "Accept" recommendation.

Reviewer #1: (No Response)

Reviewer #3: (No Response)

2. Is the manuscript technically sound, and do the data support the conclusions?

Reviewer #1: Yes

Reviewer #3: Yes

3. Has the statistical analysis been performed appropriately and rigorously? 

Reviewer #1: Yes

Reviewer #3: Yes

4. Have the authors made all data underlying the findings in their manuscript fully available?

Reviewer #1: Yes

Reviewer #3: No

5. Is the manuscript presented in an intelligible fashion and written in standard English?

Reviewer #1: Yes

Reviewer #3: Yes

6. Review Comments to the Author

Reviewer #1: (No Response)

Reviewer #3: The paper addressed important issue as it relates to Kingdom Saudi Arabia (KSA).

Below are my comments.

1. Though, the introduction looks well caved, there are more to be done to make the paper suit publication in a journal of the like of PLOS ONE. First, the authors need to be more factual and coherent in the presentation of their arguments. For instance, in making reference to the historical trend in COVID-19 outbreak globally, reference was given to January 18, 2021 while similar instance in the case of KSA referred to October 02, 2020. There is need to either update KS to 2021 or refer global data to October, 2020. The former suggestion is however preferred. See page 3 paragraph 1 lines 1-4.

2. More so, the authors need to strengthen the contribution of their work to the literature because I am aware a number of papers have been published on the topic and in KSA in particular. The arguments that the study compared several predictions using the SIR model before and after lockdown measures are the enough.

3. The paper in its present form is deficient in terms of contribution to the literature due to its lack deep review in recent studies relating to lockdown. The following studies should be considered.

https://doi.org/10.1016/j.imu.2020.100420

https://doi.org/10.1016/j.hlpt.2020.09.004

https://doi.org/10.1007/s11869-020-00918-3

https://doi.org/10.1007/s41996-020-00070-1

https://doi.org/10.1080/00036846.2020.1828809

https://doi.org/10.1016/j.trip.2020.100217

4. For robustness sake, the study can dwell more on lockdown data available at Google mobility data (https://www.google.com/covid19/mobility/). The data shows how visits to places, such as grocery stores and parks, are changing in each geographic region during the lockdown periods.

5. The language structure of the paper can be improved. For instance, the sentence structure in paragraph 2, lines 4-11 is too long. Also, when introducing new concept, full meaning should be provided before abbreviating. For instance, ICU used in paragraph 2 line 12 should be corrected and other related ones.

7. PLOS authors have the option to publish the peer review history of their article (what does this mean?). If published, this will include your full peer review and any attached files.

Reviewer #1: No

Reviewer #3: No

---

## [Author Response · Author response to Decision Letter 1]

3 Aug 2021

To Editor: We have incorporated all possible suggestions made by the reviewer. Thank you for giving us the opportunity to resubmit

Here are the reviewer's comments and our response 

Comment: 

1. Though, the introduction looks well caved, there are more to be done to make the paper suit publication in a journal of the like of PLOS ONE. First, the authors need to be more factual and coherent in the presentation of their arguments. For instance, in making reference to the historical trend in COVID-19 outbreak globally, reference was given to January 18, 2021 while similar instance in the case of KSA referred to October 02, 2020. There is need to either update KS to 2021 or refer global data to October, 2020. The former suggestion is however preferred. See page 3 paragraph 1 lines 1-4.

Response: Revised and updated the KSA data to January 18, 2021.

Comment: 

2. More so, the authors need to strengthen the contribution of their work to the literature because I am aware a number of papers have been published on the topic and in KSA in particular. The arguments that the study compared several predictions using the SIR model before and after lockdown measures are the enough.

Response: Thank you for providing suggestions for more articles. Literature review part is updated. 

Comment

3. The paper in its present form is deficient in terms of contribution to the literature due to its lack deep review in recent studies relating to lockdown. The following studies should be considered.

https://doi.org/10.1016/j.imu.2020.100420

https://doi.org/10.1016/j.hlpt.2020.09.004

https://doi.org/10.1007/s11869-020-00918-3

https://doi.org/10.1007/s41996-020-00070-1

https://doi.org/10.1080/00036846.2020.1828809

https://doi.org/10.1016/j.trip.2020.100217

Response: All the literatures mentioned here are very relevant and hence included in the literature review. Thank you for suggesting these references. 

Comment: 

4. For robustness sake, the study can dwell more on lockdown data available at Google mobility data (https://www.google.com/covid19/mobility/). The data shows how visits to places, such as grocery stores and parks, are changing in each geographic region during the lockdown periods.

Response: good suggestions. However, it is beyond the scope of the current exercise. However, the limitation is acknowledged in the discussion section. 

Comment

5. The language structure of the paper can be improved. For instance, the sentence structure in paragraph 2, lines 4-11 is too long. Also, when introducing new concept, full meaning should be provided before abbreviating. For instance, ICU used in paragraph 2 line 12 should be corrected and other related ones.

Response: Revised as suggested. Long sentence has been broken down to two sentences. Details on ICU is provided before using it as abbreviation.

---

## [Decision Letter · Decision Letter 2]

20 Aug 2021

Effect of easing lockdown and restriction measures on COVID-19 epidemic projection: a case study of Saudi Arabia

PONE-D-21-01867R2

Dear Dr. Shimul,

We’re pleased to inform you that your manuscript has been judged scientifically suitable for publication and will be formally accepted for publication once it meets all outstanding technical requirements.

Kind regards,

Siew Ann Cheong, Ph.D.

Academic Editor

PLOS ONE

Additional Editor Comments (optional):

Reviewers' comments:

Reviewer's Responses to Questions

**Comments to the Author**

1. If the authors have adequately addressed your comments raised in a previous round of review and you feel that this manuscript is now acceptable for publication, you may indicate that here to bypass the “Comments to the Author” section, enter your conflict of interest statement in the “Confidential to Editor” section, and submit your "Accept" recommendation.

Reviewer #3: All comments have been addressed

2. Is the manuscript technically sound, and do the data support the conclusions?

Reviewer #3: Yes

3. Has the statistical analysis been performed appropriately and rigorously? 

Reviewer #3: Yes

4. Have the authors made all data underlying the findings in their manuscript fully available?

Reviewer #3: Yes

5. Is the manuscript presented in an intelligible fashion and written in standard English?

Reviewer #3: Yes

6. Review Comments to the Author

Reviewer #3: I congratulate the authors for a job well done. They have provided sufficient review in lieu of the comments raised.

7. PLOS authors have the option to publish the peer review history of their article (what does this mean?). If published, this will include your full peer review and any attached files.

Reviewer #3: No

---

## [Editor Report · Acceptance letter]

31 Aug 2021

PONE-D-21-01867R2 

Effect of easing lockdown and restriction measures on COVID-19 epidemic projection: a case study of Saudi Arabia 

Dear Dr. Shimul:

I'm pleased to inform you that your manuscript has been deemed suitable for publication in PLOS ONE. Congratulations! Your manuscript is now with our production department. 

Kind regards, 

on behalf of

Dr. Siew Ann Cheong 

Academic Editor

PLOS ONE